# Thermochemistry, Tautomerism, and Thermal Stability of 5,7-Dinitrobenzotriazoles

**DOI:** 10.3390/ijms24065330

**Published:** 2023-03-10

**Authors:** Igor N. Melnikov, Vitaly G. Kiselev, Igor L. Dalinger, Alexey M. Starosotnikov, Nikita V. Muravyev, Alla N. Pivkina

**Affiliations:** 1Semenov Federal Research Center for Chemical Physics RAS, 4 Kosygina Str., 119991 Moscow, Russia; 2Physics Department, Novosibirsk State University, 1 Pirogova Str., 630090 Novosibirsk, Russia; 3Institute of Chemical Kinetics and Combustion SB RAS, 3 Institutskaya Str., 630090 Novosibirsk, Russia; 4Zelinsky Institute of Organic Chemistry RAS, 47 Leninsky Ave., 119991 Moscow, Russia

**Keywords:** thermal stability, thermokinetic analysis, calorimetry, quantum chemical calculations, decomposition mechanism

## Abstract

Nitro derivatives of benzotriazoles are safe energetic materials with remarkable thermal stability. In the present study, we report on the kinetics and mechanism of thermal decomposition for 5,7-dinitrobenzotriazole (**DBT**) and 4-amino-5,7-dinitrobenzotriazole (**ADBT**). The pressure differential scanning calorimetry was employed to study the decomposition kinetics of **DBT** experimentally because the measurements under atmospheric pressure are disturbed by competing evaporation. The thermolysis of **DBT** in the melt is described by a kinetic scheme with two global reactions. The first stage is a strong autocatalytic process that includes the first-order reaction (*E*_a1_^I^ = 173.9 ± 0.9 kJ mol^−1^, *log*(*A*_1_^I^/s^−^^1^) = 12.82 ± 0.09) and the catalytic reaction of the second order with *E*_a2_^I^ = 136.5 ± 0.8 kJ mol^−1^, *log*(*A*_2_^I^/s^−^^1^) = 11.04 ± 0.07. The experimental study was complemented by predictive quantum chemical calculations (DLPNO-CCSD(T)). The calculations reveal that the 1H tautomer is the most energetically preferable form for both **DBT** and **ADBT**. Theory suggests the same decomposition mechanisms for **DBT** and **ADBT**, with the most favorable channels being nitro-nitrite isomerization and C–NO_2_ bond cleavage. The former channel has lower activation barriers (267 and 276 kJ mol^−1^ for **DBT** and **ADBT**, respectively) and dominates at lower temperatures. At the same time, due to the higher preexponential factor, the radical bond cleavage, with reaction enthalpies of 298 and 320 kJ mol^−1^, dominates in the experimental temperature range for both **DBT** and **ADBT**. In line with the theoretical predictions of C–NO_2_ bond energies, **ADBT** is more thermally stable than **DBT**. We also determined a reliable and mutually consistent set of thermochemical values for **DBT** and **ADBT** by combining the theoretically calculated (W1-F12 multilevel procedure) gas-phase enthalpies of formation and experimentally measured sublimation enthalpies.

## 1. Introduction

Benzotriazoles are an important class of heterocyclic compounds with a wide range of applications. For instance, they are employed as important intermediates in organic synthesis [1,2], biologically active agents in medicine and agriculture [3,4,5], and inhibitors of metal corrosion [6,7,8]. Nitro derivatives of benzotriazoles are also considered as promising energetic materials with generally low sensitivity to mechanical stimuli and high thermal stability [9,10,11]. Two representative benzotriazoles, namely, 5,7-dinitrobenzotriazole (**DBT**) and 4-amino-5,7-dinitrobenzotriazole (**ADBT**), are studied in the present work (Figure 1). While the first reports on **DBT** date back to 1897 [12] and several new synthetical procedures have been proposed since that time [13,14,15], **ADBT** has been synthesized for the first time only recently [16]. Both compounds have a moderate density of ~1.66 g cm^−3^ and, consequently, mediocre energetic performance (calculated detonation velocities ~7.3 km s^−1^ [16]). However, these materials have remarkably high thermal stability, as confirmed in linear heating tests by differential scanning calorimetry (DSC). **DBT** melts at 466 K (193 °C) and then decomposes with the extrapolated onset of 564 K (291 °C), while **ADBT** melts with a subsequent decomposition at 610 K (337 °C) [16]. These values are the only available information on the thermal behavior of **DBT** and **ADBT**. At the same time, benzotriazoles with various non-explosophoric moieties have been extensively studied, and their pyrolysis and photolysis were proposed to occur via N_2_ elimination [17,18,19,20]. In order to obtain a deeper insight into the possible decomposition mechanisms of dinitrobenzotriazoles, we also considered the structurally similar dinitrotriazolopyridines **3** and **4**, and simpler species with the dinitrobenzotriazoles skeleton, viz., 1,3-nitrobenzene (**5**) and 1,2,3-triazole (**6**, Figure 1). Recently, we have studied the thermal stability of the similar compounds **3** and **4** using pressure DSC and quantum chemical calculations at the DLPNO-CCSD(T) level [21]. We found that the thermolysis of **3** occurs in the melt and is described by a kinetic scheme with two autocatalytic independent parallel reactions, whereas the decomposition kinetics of **4** is more complex due to its overlap with the melting. Theoretical calculations revealed the initial decomposition channel to change from the nitro-nitrite rearrangement (*E*_a_~240 kJ mol^−1^) to the C–NO_2_ bond cleavage with an activation barrier of ~280 kJ mol^−1^ at temperatures around 500 K.

The thermolysis of the prototypical 1,3-dinitrobenzene (**5**) was investigated in the gas phase by the manometry technique [22] and laser-powered homogeneous pyrolysis [23]. In both studies, the thermolysis kinetics of **5** was described as a first-order process with similar values of the Arrhenius parameters *E*_a_ = 285 kJ mol^−1^, *log*(*A*/s^−1^) = 16.9 [22] and *E*_a_ = 293 kJ mol^−1^, *log*(*A*/s^−1^) = 14.5 [23]. The radical C–NO_2_ bond cleavage was hypothesized to be a rate-limiting process. Shao et al. [24] reported the theoretical estimations of the C-NO_2_ bond dissociation energy for **5** to be 267–285 kJ mol^−1^ at various DFT levels of theory. For the related nitro-amino compounds, a wider set of possible primary decomposition pathways has been proposed. For instance, in the case of TATB (1,3,5-triaminotrinitrobenzene) several particular initial pathways were discussed in the literature. Apart from the conventional C–NO_2_ bond scission and nitro-nitrite isomerization, DFT calculations were used to consider intra- and intermolecular hydrogen transfers and intramolecular cyclization forming benzofurazan or benzofuroxan derivatives [25,26].

The mechanism of 1,2,3-triazole thermolysis has been reported in several theoretical studies. Da Silva et al. [27] showed at the G3B3 level that the ring opening with the subsequent N_2_ elimination with the activation energy of 222 kJ mol^−1^ dominates the thermolysis of 1,2,3-triazole. Recently, Lu et al. [28] has refined the activation barrier of this reaction using the DLPNO-CCSD(T) procedure and obtained the value of ~189 kJ mol^−1^.

It is also worth mentioning that **DBT** and **ADBT** are prone to annular tautomerism. The molecular structure of N-unsubstituted benzotriazole has been the subject of numerous structural studies [29,30,31]. Recently, Santa Maria et al. [32] used X-ray diffraction, NMR spectroscopy, and DFT calculations to demonstrate that the 5,6-dinitrobenzotriazole exists in the solid state as the *1H* tautomer (**7**, Figure 1), while the *2H* tautomer (**8**, Figure 1) is energetically more favorable in the gas phase. In the DMSO solution, the mixture of both tautomers coexists. Similarly, for a long time, the molecular structure of **DBT** was attributed to that of 4,6-dinitrobenzotriazole (denoted as *1H-4,6-*tautomer henceforth) [12,13,14]. However, Graham et al. [15] employed B3LYP calculations to show that the 5,7-dinitro-1H-benzotriazole (*1H-5,7-*tautomer) isomer is ~35 kJ mol^−1^ more favorable than the *1H-4,6* isomer in the gas phase. Then, Ehler et al. [16] experimentally confirmed the molecular structure of **DBT** using X-ray diffraction data. Moreover, the authors reported the structure of **ADBT**, which was also attributed to *1H-5,7-*tautomer. At the same time, the role of tautomeric interconversions of benzotriazoles in their decomposition mechanism remains unclear.

The literature survey reveals that the thermal behavior and decomposition mechanisms of nitro-substituted benzotriazoles are not well understood. Thus, in the present contribution, we address the thermochemistry, tautomerism, and thermal stability (kinetics and decomposition mechanism) of **DBT** using thermal analysis techniques and quantum chemical calculations. To avoid evaporation, the experiments were performed using pressure differential scanning calorimetry (DSC), and then the kinetic data were fitted with advanced thermokinetic methods, e.g., isoconversional and model-fitting approaches. The experiment was complemented by high-level reliable quantum chemical calculations of the primary decomposition mechanism of **DBT** at the DLPNO-CCSD(T) level of theory. By analogy with the well-known energetic materials bearing similar fragments, we suggest that the primary decomposition mechanism of the dinitrobenzotriazoles involves the C–NO_2_ radical bond cleavage and the molecular isomerizations, either nitro-nitrite or H-transfers yielding the aci-form. Moreover, we studied the influence of the amino group on the thermal stability of the dinitrobenzotriazole skeleton by considering the primary decomposition pathways of **ADBT**. 

## 2. Results and Discussion

### 2.1. Thermal Behaviour of DBT under Atmospheric Pressure

Simultaneous thermal analysis data for **DBT** heated at a 1 K min^−1^ rate at atmospheric pressure are shown in Figure 2. The DSC curve (blue trace, Figure 2) shows two endothermic effects. The first corresponds to the melting of a sample. The melting temperature determined as the extrapolated peak onset is 475.0 K. Note that this value is remarkably higher than those reported previously, viz., 466, 471, and 469–470 K [12,13,15,16]. These discrepancies can be attributed to higher sample purity in the present study or insufficient temperature control in the previous experiments. The second endothermic effect corresponds to evaporation, which is accompanied by the complete mass loss of the sample (red trace, Figure 2). Hence, to observe the exothermic effect of the thermal decomposition, we applied pressure to be DSC (PDSC), which is under elevated external pressure. This approach allowed us to shift the evaporation to higher temperatures and very often observe the sole decomposition kinetics of a compound under study [33,34].

### 2.2. Thermal Kinetics of DBT under Elevated Pressure

The pressure DSC experiments were performed under an elevated external pressure of 2.0 MPa and at heating rates varying from 0.5 to 10 K min^−1^. Thermal decomposition of **DBT** occurs in the melt, and the PDSC curves have the characteristic shape with the two exothermic peaks (Figure 3). The first peak (*I*) has a sharp form, whereas the second (*II*) has a lower amplitude without a visible maximum and apparently corresponds to the heat released in secondary reactions. Note that peak *II* exhibits a broad and gradual decrease to the baseline. This effect can introduce inaccuracies in determining the total heat effect and subsequent thermokinetic modeling. It is worth mentioning that **DBT** exhibits very similar thermal behavior to its structural analog, 6,8-dinitrotriazolopyridine (**3**) [21]. The decomposition heat effect and residual sample mass dependences on the heating rate can be found in the Appendix A.

The rough estimations of **DBT** decomposition kinetics were performed by the Kissinger method applied to peak *I* of the DSC data (Figure 3 and Figure 4). The Kissinger method yields the activation energy and the preexponential factor to be *E*_a_ = 157 ± 1 kJ mol^−1^ and *log*(*A*/s^−1^) = 11.8 ± 0.1. These values were used as an initial guess for the first reaction in the model-fitting analysis. At the same time, the second exothermic peak *II* does not exhibit a clear maximum within the range of heating rates applied. Thus, the Kissinger method is not directly applicable in this case.

Then, a more detailed description of the decomposition kinetics of **DBT** was obtained by isoconversional Friedman analysis, which tracks the change of activation energy *E*_a_ against a conversion degree *α*. The isoconversional plot in Figure 5 shows the considerable variation of activation energy throughout the process, with the two regions with relatively constant activation energy, which may indicate the activation barriers for contributing reaction steps [35]. At the beginning of decomposition (α < 0.2), the activation energy stabilizes around 146 ± 2 kJ mol^−1^ (Figure 5), which slightly differs from the output of the Kissinger approach for peak *I* (cf. Figure 4). The reason for this is that the maximum of the DSC peak corresponds to larger conversion degree values α = 0.25 ± 0.02. Upon the reaction’s progress, the activation energy decreases and reaches the minimum value ~96 kJ mol^−1^ at α = 0.34, possibly due to competition between the different reactions. Next, at the conversion degrees 0.55 < α < 0.75, the isoconversional plot shows the second region with the constant activation energy, which approximately coincides with the low conversion degree value *E*_a_ = 146 ± 1 kJ mol^−1^. After that, the activation energy rises to 169 kJ mol^−1^ at the end of the process (Figure 5). 

Next, we applied the model-fitting approach to analyze the complex decomposition behavior of **DBT**. From the shape of DSC curves and results of isoconversional analysis, we considered several two-step kinetic schemes, viz., independent parallel, consecutive, and competitive parallel processes. These reactions obey a model of the extended Prout-Tompkins Equation (6). According to the Bayesian information criteria (the lower values correspond to a better model), among the two-step models, the best fit of the experimental data offers the kinetic scheme with two consecutive reactions (Appendix A). The first reaction has the following kinetic parameters in Equations (3) and (6): *E*_a1_ = 153.7 ± 0.3 kJ mol^−1^, *log*(*A*_1_/s^−1^) = 12.56 ± 0.02, *n*_1_ = 0.99 ± 0.03, *m*_1_ = 1.95 ± 0.02, and *q*_1_ = 0.9. The optimized values of the exponents *n*_1_ and *m*_1_ are close to integer numbers. Hence, the first reaction model is close to the autocatalytic process with a second order by the autocatalyst. The second global reaction is more complex in terms of the effective kinetic parameters: *E*_a2_ = 90.9 ± 2.0 kJ mol^−1^, *log*(*A*_2_/s^−1^) = 5.6 ± 0.2, *n*_2_ = 1.28 ± 0.05, *m*_2_= –0.31 ± 0.01, and *q*_2_ = 0.9 (Table 1, model C).

Since the first reaction of **DBT** thermolysis in the melt exhibits an autocatalytic nature, it is also reasonable to probe the conventional autocatalytic model [36]. Thus, the final formal scheme includes the first global reaction step, which is, in turn, comprised of two parallel reactions, namely, the first-order process and an *m*-th order autocatalytic reaction (AC):(1)$$\frac{d\alpha }{dt}=k_{1}\left(1-\alpha \right)+k_{2}\alpha^{m}\left(1-\alpha \right),$$
while the second global step model remains in the ePT form of Equation (6). Actually, Equation (1) is a particular case of the model with two parallel reactions in the ePT form (6) with *q*_1_ = *q*_2_ = 1 and *m*_1_ = 0. In the present case, we employed the integer values of *m* = 1 and 2.

The model optimization revealed that the modified kinetic scheme with *m* = 2 improves the fit quality and its statistical performance (Table 1, Model E, and Figure 6). Note that the reaction rate of the autocatalytic part of the first global reaction is two orders of magnitude faster than its non-catalytic counterpart ($k_{1}/k_{2}\approx 59$, Figure 6). The final kinetic parameters for the noncatalytic decomposition reaction in the melt are: *E*_a1_^I^ = 173.9 ± 0.9 kJ mol^−1^, *log*(*A*_1_^I^/s^−1^) = 12.82 ± 0.09.

### 2.3. Theoretical Calculations

As seen from the previous section, the non-isothermal thermoanalytical experiment yielded only the effective kinetic parameters of the thermal decomposition of **DBT** without any mechanistic assumptions. It is impossible to separate the contributions from primary and secondary elementary chemical reactions. In such cases, quantum chemical calculations often complement the experiment in a very effective way. We applied modern quantitative quantum chemistry approaches to calculating the activation barriers and rate constants of the primary decomposition reactions of **DBT**. Furthermore, we investigated the influence of the introduction of the amino group in the benzotriazole skeleton on thermal stability, considering the primary decomposition reactions of the **ADBT** compound as well.

#### 2.3.1. Mutual Interconversions of DBT and ADBT Tautomers

We started with the analysis of the molecular structure of benzotriazoles. We considered three different annular tautomeric forms, viz., *1H-5,7-tautomer*, *2H-4,6-tautomer*, and *1H-4,6-tautomer* (Figure 7). The single unimolecular hydrogen transfer via **_x_TS1** (where **x** corresponds to **DBT** and **ADBT**, respectively) leads to *2H-4,6-tautomer* with a high activation barrier ~235 kJ mol^−1^, whereas the subsequent hydrogen transfer via **_x_TS2** with a similar barrier ~240 kJ mol^−1^ gives the *1H-4,6-tautomer* (Figure 7). Note that the *1H-5,7*-form lies significantly lower on the PES, viz., by ~25–30 kJ mol^−1^ than *2H-4,6* and by ~30–56 kJ mol^−1^ compared to the *1H-4,6-tautomer*. Thus, the *1H-5,7-tautomer* of **DBT** and **ADBT** is the most energetically preferable one. These results agree well with previous DFT calculations for **DBT [15]** and also with the crystalline structures for both compounds from X-ray diffraction experiments [16]. Therefore, for all following calculations of decomposition pathways, the *1H-5,7-form* was chosen as a reference species.

The tautomeric interconversions can also proceed via bimolecular reactions in the dimers. As shown in our previous works, the concerted double hydrogen transfer in the dimers often occurs with lower activation barriers than the corresponding unimolecular reactions [37,38]. In the present case, we considered the concerted double hydrogen transfer in the dimers from the *1H-5,7-tautomer* yielding the *2H-4,6* and *1H-4,6* forms (Figure 8). It was found that the activation barriers of **_x_TS_D12_** and **_x_TS_D34_** are significantly lower than those for unimolecular processes. Consequently, if the tautomeric interconversions are faster than the unimolecular decomposition pathways, then the tautomeric forms are in equilibrium during the thermolysis of **DBT** and **ADBT**. Therefore, the *2H-4,6* and *1H-4,6* forms can also contribute to the primary decomposition mechanism. Thus, we consider next the monomolecular decomposition reactions of the **DBT** and **ADBT** tautomers.

#### 2.3.2. Mechanisms of the Primary Decomposition Reactions of DBT and ADBT

We examined the primary decomposition mechanisms proposed in the literature for structurally similar nitroaromatic compounds [25,26,39]: the radical C–NO_2_ bond cleavage, nitro-nitrite rearrangement, the reactions involving the aci intermediate, and the triazole ring opening channel, which is widely discussed for 1, 2, 3-triazole and benzotriazoles [18,19,27,28]. For these channels, all non-equivalent positions of NO_2_ groups and hydrogen atoms were considered.

Let us start with the decomposition mechanisms for the *1H-5,7-tautomers* of **DBT** and **ADBT**. The barrierless radical C-NO_2_ bond cleavage yields heterocyclic radicals •**_x_R6** and •NO_2_ (Figure 9). The corresponding reaction enthalpies are 299.2 and 318.7 kJ mol^−1^ for **DBT** and **ADBT**, respectively. These enthalpies agree well with the typical values for nitroaromatic compounds, including 1,3-dinitrobenzene (**5**) [22,23,40,41]. 

The activation barriers of a competitive nitro-nitrite rearrangement **_x_TS7** are 32 and 42 kJ mol^−1^ lower than the corresponding radical asymptotes, **•_x_R6** + •NO_2_ (Figure 9), respectively. The nitrite species **_x_P7**, formed in primary reactions, are prone to very fast elimination of the •NO radical, with reaction enthalpies of 80 and 58 kJ mol^−1^ for **DBT** and **ADBT**, respectively (Figure 9). 

The N_2_ elimination channel proceeds via two subsequent elementary reactions. The triazole ring opening occurs first (**xTS8**), yielding the diazo intermediate **xP8**, which has a low activation barrier of reverse reaction E_a_~10 kJ mol^−1^, whereas the second **xTS8a** leads to the release of N_2_ and the formation of **xP8a** (Figure 9). Since the reaction via the transition state **xTS8** is reversible, the kinetics of the N_2_ elimination is determined by the effective activation barrier of **xTS8a**. The corresponding values of 281.4 and 286.4 kJ mol^−1^ for **DBT** and **ADBT,** respectively, are slightly higher (in 10–15 kJ mol^−1^) than those for the nitro-nitrite rearrangement **_x_TS7** and lower than the enthalpy of C-NO_2_ bond cleavage by 32 and 18 kJ mol^−1^ (Figure 9). Note that these values largely exceed their counterparts for unsubstituted 1,2,3-triazole (**6**) by 60 [27] and 100 kJ mol^−1^ [28]. 

In addition, we considered the reactions of intramolecular isomerization to aci-form via **xTS3** and **xTS4** and the hydrogen transfer from the benzene ring to a triazole moiety via **xTS5** (Appendix A). The aci-isomerizations **xTS3** and **_ADBT_TS4** have activation barriers 40–80 kJ mol^−1^ lower than those of tautomeric interconversions. However, the activation barriers of the reverse reactions *E*a~10–25 kJ mol^−1^ indicate that the intermediates **xP3** and **xP4** should rearrange fast to the initial reagent. Furthermore, the hydrogen transfer **xTS5** with the activation barriers of 394 kJ mol^−1^ for **DBT** and 196 kJ mol^−1^ for **ADBT** is the least favorable process among the isomerization pathways (Appendix A). Among the considered unimolecular isomerizations, only the aci-form **xP3** and **_ADBT_P4** can take part in the subsequent decomposition. We considered the radical elimination of •NO_2_, •OH, and the molecular elimination of HONO. These channels have effective activation barriers higher than 320 kJ mol^−1^ (the details can be found in the Appendix A); therefore, they are kinetically unimportant.

In the case of tautomers *2H-4,6* and *1H-4,6*, we examined the most competitive decomposition pathways, i.e., the C-NO_2_ bond cleavage and the nitro-nitrite rearrangement (Appendix A). We found that the relative reaction enthalpy for radical bond cleavage exceeds 15–30 kJ mol^−1^ for *1H-5,7-tautomers*. The nitro-nitrite rearrangements have closer activation barriers to the *1H-5,7-tautomer* for **DBT** (difference ~1–3 kJ mol^−1^), but for **ADBT** the difference is still large. Overall, the lowest activation barriers correspond to *1H-5,7-form* (**_x_TS7** and **•_x_R6** + •NO_2_, Figure 9). These results suggest that thermolysis of dinitrobenzotriazoles occurs via *1H-5,7-tautomer*.

The initial decomposition reactions of **DBT** and **ADBT** comprise the same channels (C-NO_2_ bond cleavage, nitro-nitrite rearrangement, and N_2_ elimination). As seen in Figure 9, the activation barriers for amino-substituted dinitrobenzotriazole are noticeably higher, which implies its higher thermal stability as compared to **DBT**. Therefore, the introduction of the amino group to the benzotriazole moiety increases its thermal stability. This fact is in line with our previous results for dinitrotriazolopyridines **3** and **4** [21]. 

In the experiment, the thermolysis of **DBT** proceeds in a melt. According to the literature [16], the thermal decomposition of **ADBT** also occurs in the melt at heating rates higher than 5 K min^−1^. To estimate the influence of the media on the primary decomposition reactions of **DBT** and **ADBT**, we used the PCM model. Several model solutions, viz., the isotropic media with ε = 1.96 (cyclopentane), ε = 19.26 (2-propanol), and ε = 46.83 (dimethyl sulfoxide), were used in the PCM calculations. The latter value is exceptionally high and was used to demonstrate the insensitivity of the thermolysis mechanisms of **DBT** and **ADBT** to the dielectric properties of the melt. Indeed, the results of PCM calculations show only marginal changes for important decomposition channels (further details can be found in the Appendix A). Note that the tautomers *2H-4,6* and *1H-4,6* become more thermally accessible in the solutions with high polarity (2-propanol and DMSO); however, the dominant role of the *1H-5,7-tautomer* persists. Thus, we believe the C-NO_2_ bond cleavage, nitro-nitrite rearrangement, and N_2_ elimination remain competitive processes in the melt. 

To determine the dominant channel among the primary decomposition pathways, we calculated the rate constants of the elementary reactions of **DBT** and **ADBT** using the transition state theory (Equation (7)). In the case of barrierless reactions, the phase space theory (Equation (9)) was employed to localize the transition state. Note that for the decomposition pathways with a reversible first step, e.g., the **xTS8**, the effective rate constant was calculated as $k_{eff}≅\frac{k_{1}}{k_{-1}}k_{lim}$, where $\frac{k_{1}}{k_{-1}}$ is the equilibrium constant of the first step (*k*_1_ is the rate constant of a direct reaction, *k*_-1_-the rate constant of the reversion), and *k*_lim_ is the rate constant of the limiting process occurring via **xTS8a**. The Arrhenius parameters for all decomposition channels are given in the Appendix A. The rate constants for the most important primary decomposition pathways are summarized in Table 2 and Figure 10. 

From the energetic point of view, the nitro-nitrite rearrangements **_x_TS7** are the most favorable primary channels, followed by the two-step N_2_ elimination via **_x_TS8**, and by the (least energetically preferred of three) C-NO_2_ bond cleavage yielding **•_x_R6** + •NO_2_ (Figure 9). On the other hand, the radical channels have the largest preexponential factors (Table 2) typical of barrierless reactions [40,41]. The latter values exceed by two and five orders of magnitude the more energetically favorable channels, **_x_TS8** and **_x_TS7**, respectively (Table 2). Figure 10 compares the rate constants for the competitive primary channels extrapolated by the Arrhenius equation (Equation 6). As seen from Figure 10, the N_2_ elimination via **_x_TS8** is overall the least favorable process, while the nitro-nitrite rearrangement and C-NO_2_ bond cleavage have the closest rate constants. Moreover, the high reaction enthalpy of **•_x_R6** + •NO_2_ pathway is compensated by a greater preexponential factor (or, equivalently, the activation entropy), which leads to the change of the dominant channel at relatively low temperatures (viz., 307 K for **DBT** and 395 K for **ADBT**) from the nitro-nitrite isomerization (red lines, Figure 10) to the radical C-NO_2_ bond cleavage (blue graphs, Figure 10). Taking into account the temperature range where the decomposition was observed experimentally (Figure 2 and Figure 3), and the results of PCM calculations, we propose that the thermolysis of both **DBT** and **ADBT** in the melt occurs via the radical C-NO_2_ bond cleavage.

It is also instructive to compare the thermal stability of the benzotriazoles compounds with their structural isomers **3** and **4**. Since the experimental kinetics for all compounds is very complicated, we compared the results of theoretical calculations. Quantum chemical calculations of the effective rate constants for the C–NO_2_ bond scission at the mean experimental temperature of 650 K are presented in Table 3. These results indicate that the dinitrobenzotriazoles **DBT** and **ADBT** have higher thermal stability than the dinitrotriazolopyridines **3** and **4**. 

#### 2.3.3. Thermochemistry of the Compounds Studied

The gas-phase enthalpy of formation ($\Delta_{f}H_{m}^{{}^{\circ}}(g)$) for dinitrobenzotriazoles and their structural analogs, dinitrotriazolopyridines **3** and **4**, were calculated using the high-level computational W1-F12 method. The sublimation enthalpies ($\Delta_{s}^{g}H_{m}^{{}^{\circ}}$) are necessary to complement the gas-phase $\Delta_{f}H_{m}^{{}^{\circ}}(g)$ values to obtain the solid-state enthalpy of formation $\Delta_{f}H_{m}^{{}^{\circ}}(s)$. In the case of **DBT**, the sublimation enthalpy was determined experimentally using thermogravimetry at atmospheric pressure [42] (Appendix A). The higher temperature of the region used to derive the vaporization data at every heating rate was smaller than the onset of the rise of PDSC curves at the same rate. Hence, the thermal decomposition was supposed to not influence the results. The evaporation enthalpy value corrected to room temperature ($\Delta_{l}^{g}H_{m}^{{}^{\circ}}$= 94.6 ± 1.9 kJ mol^−1^), was supplemented by the melting enthalpy from pressure DSC tests ($\Delta_{s}^{l}H_{m}^{{}^{\circ}}$= 27.3 ± 1.3 kJ mol^−1^) to give the final value $\Delta_{s}^{g}H_{m}^{{}^{\circ}}$= 121.8 ± 3.2 kJ mol^−1^.

For **ADBT** and dinitrotriazolopyridines, the sublimation enthalpies were estimated using the recently suggested modified empirical Trouton-Williams rule for CHNO compounds [42]:(2)$$\Delta_{s}^{g}H_{m}^{0}=0.15\cdot T_{m}+3.27\cdot \left[H\right]+5.30\cdot \left[N\right]+3.30\cdot \left[O\right].$$

Note that the outcome of Equation (2) matches the experimental result within the experimental error (Table 4). Then, the solid-state enthalpy of formation was calculated according to the Hess law, $\Delta_{f}H_{m}^{{}^{\circ}}\left(s\right)=\Delta_{f}H_{m}^{{}^{\circ}}\left(g\right)-\Delta_{s}^{g}H_{m}^{{}^{\circ}}$. The obtained results are displayed in Table 4 against available literature data, which comprise only the theoretical estimates. Considered structural isomers pairwise (**DBT**–**3** and **ADBT**–**4**) show very close thermochemistry parameters (Table 4). The calculated values of $\Delta_{f}H_{m}^{{}^{\circ}}(s)$ in the present study differ from the literature data. However, the methodology used in our work is generally more reliable.

## 3. Methods and Materials

### 3.1. Materials

5,7-Dinitrobenzotriazole (**DBT**) was synthesized according to the modified literature procedure [16]. A mixture of 70% HNO_3_ (23 mL) and concentrated H_2_SO_4_ (23 mL) was added dropwise to the suspension of benzotriazole (20 g, 0.168 mol) in H_2_SO_4_ (80 mL) at 15–20 °C. After stirring for 24 h at 20 °C, the mixture was poured into ice water; a precipitate was filtered off, washed with water, and dried on air to give 4-nitrobenzotriazole (25.5 g, 92.5%), which was used without further purification. Crude 4-nitrobenzotriazole (10 g, 47.8 mmol) was suspended in concentrated H_2_SO_4_ (120 mL), cooled to 0 °C, and then fuming HNO_3_ (20 mL, d 1.5 g/cm^3^) was added dropwise. A reaction mixture was stirred for 1 h at 20 °C and for 3 h at 120 °C. After cooling to 20 °C, it was poured into the crushed ice. The resulting precipitate was filtered off, thoroughly washed with H_2_O, dried on air, and recrystallized from EtOH to give 5,7-dinitrobenzotriazole (7.7 g, 57%) as a white solid. ^1^H NMR (DMSO-d_6_): 9.00 (s, 1H); 9.50 (s, 1H). ^13^C NMR (DMSO-d_6_): 118.9, 123.2, 129.9, 133.2, 143.6, 146.5. HRMS (ESI), found: *m*/*z* 210.0257 [M+H]; calculated for C_6_H_3_N_5_O_4_: 210.0258. Elemental analysis calculated (%) for C_6_H_3_N_5_O_4_: C, 34.46; H, 1.45; N, 33.49; found: C, 34.57; H, 1.39; N, 33.60.

### 3.2. Thermoanalytical Experiments

The thermal behavior of **DBT** was examined by the simultaneous thermal analyzer STA 449 F1 (Netzsch), which combines thermogravimetry (TG) and differential scanning calorimetry (DSC). The samples of ca. 1.8 mg weight were placed in open aluminum crucibles and heated at a rate *β* = 10 K min^−1^ under an argon flow of 50 mL min^−1^.

The thermal decomposition of **DBT** was assessed using the pressure differential scanning calorimeter DSC 204 HP (Netzsch). Samples were poured into closed aluminum crucibles with pierced lids and heated with the rates *β* of 0.5, 1, 2, 5, and 10 K min^−1^ up to 500 °C. The instrument has been calibrated with respect to heat flow and temperature using highly-pure metal calibrants, and all measurements were performed under a nitrogen flow of 150 mL min^−1^. The sample mass of **DBT** was varied from 0.2 to 3.9 mg with the heating rate to keep the DSC signal magnitude below 8 mW to minimize the detrimental self-heating phenomenon [45]. 

The thermokinetic analysis of **DBT** was performed using various kinetic techniques, from the conventional Kissinger method to modern isoconversional and model-fitting approaches. The basic equation of thermokinetics is
(3)$$\frac{d\alpha }{dt}=k\left(T\right)f\left(\alpha \right),$$
where α is a conversion degree, $k\left(T\right)$ is an Arrhenius dependence of the rate constant against temperature, $f\left(\alpha \right)$—the kinetic model in a differential form. For the DSC data, the conversion degree is determined as the ratio of a partial area under the DSC curve with respect to the total heat release of the reaction. The initial evaluation of the kinetic parameters was performed using the Kissinger approach [45,46], which gives an apparent activation energy *E*_a_ and a preexponential factor *A* from the shift of the DSC peak temperature *T*_p_ against a heating rate *β*:(4)$$E_{a}=-R\frac{d\ln(\beta /{T_{p}}^{2})}{d\left(1/T_{p}\right)}.$$

More detailed information on the thermal kinetics can be obtained from the isoconversional Friedman analysis [47], which yields the Arrhenius parameters *A* and *E*_a_ against the conversion degree α:(5)$${\ln\left(\frac{d\alpha }{dt}\right)}_{\alpha ,i}=\ln\left[f(\alpha )\cdot A_{\alpha }\right]-\frac{E_{a}}{RT_{\alpha ,i}},$$
where a subscript *i* corresponds to a particular measurement. To determine the final kinetic model, we employed the model fitting analysis along with the “top-down” approach [48]. To this end, we optimized the reaction model in the flexible form of an extended Prout−Tompkins (ePT) equation [49]:(6)$$f(\alpha )={\left(1-\alpha \right)}^{n}{\left[1-q\left(1-\alpha \right)\right]}^{m}.$$

Equation (6) is referred to as ePT(*n*, *m*, *q*). Initially, the single reaction was probed, and if necessary, more reaction steps were added. Additionally, the exponents in Equation (6) were fixed to particular integer values to reduce the model to one of the ideal theoretical reaction types (e.g., a first-order reaction, a one-dimensional diffusion, etc.).

All thermokinetic calculations were carried out using self-developed open-source THINKS software [50] and under ICTAC recommendations [35,51].

### 3.3. Quantum Chemical Calculations

Electronic structure calculations were carried out using Gaussian 16 [52], ORCA 4.2.1 [53,54], and Molpro 2010 [55] quantum chemical programs. The density functional theory (DFT) calculations at the M06-2X/6-311++G(2df,p) [56] level were used for the geometry optimization of all considered species (reactants, products, and transition states), as well as for calculations of frequencies and thermal corrections to thermodynamic potentials. All the stationary point and transition state structures correspond to minima on the potential energy surfaces (PES). To confirm the nature of localized transition states, the intrinsic reaction coordinate (IRC) approach [57] was used. Gaussian 16 software was used to perform all the calculations at the DFT level of theory [52]. To get the desired “chemical” accuracy of ~4 kJ mol^−1^ for the activation barriers, we refined the single-point electronic energies using the DLPNO-CCSD(T0) methodology (the “Normal PNO” truncation thresholds were used) [58] with the jun-cc-pVQZ “seasonally” augmented basis set [53]. Recent reports demonstrated that this technique can be used to obtain the accurate thermochemistry and activation barriers for the cage and heterocyclic nitramino energetic materials, such as hexanitrohexaazaisowurtzitane (CL-20) [59,60], at a reasonable computational cost. To accelerate the convergence of the SCF components of DLPNO-CCSD(T) energy, the RIJK density fitting (DF) approximation [61] was applied. For the calculations of corresponding integrals in the framework of the DF approximation, the auxiliary basis sets (aug-cc-pVQZ/JK and aug-cc-pVQZ/C in the ORCA nomenclature) [53] were used. The multireference character of the wave functions of the reagents, intermediates, and transition states was estimated using the T1 diagnostic during the DLPNO-CCSD(T) calculations [62]. Modest T1 values obtained in all cases (<0.020) indicate that a single reference-based electron correlation procedure is appropriate in the present case. All DLPNO-CCSD(T) calculations were performed using the ORCA 4.2.1 [53,54] set of programs.

The rate constants of monomolecular reactions in the gas phase in the high-pressure limit were computed in accordance with the canonical transition state theory (TST):(7)$$k\left(T\right)=\alpha \frac{kT}{h}exp\left(-\frac{\Delta G^{\neq }(T)}{kT}\right),$$
where α is a statistical factor (a number of equivalent reaction channels), and Δ*G*^≠^(*T*) is the free energy of activation calculated using the DLPNO-CCSD(T)/jun-cc-VQZ electronic energies and corresponding M06-2X thermal corrections. The TST rate constants were calculated in the temperature range 300–1000 K with a step of 100 K and then approximated by the Arrhenius equation:(8)$$k=Aexp\left(-\frac{E_{a}}{RT}\right),$$

The rate constants of barrierless radical C–NO_2_ bond cleavage were estimated using the phase space theory [63,64] implemented in the PAPR software [65]. In the framework of this theory, the interaction between the radical moieties is described with the potential:(9)$$U=-\frac{C_{n}}{r^{n}},$$
where parameters *C*_n_ = 6.0 and *n* = 5.0 were chosen to match the high-pressure limit of the rate constant of hexogen radical decomposition [66]. 

To understand the reactivity of DBT and ADBT in the melt, the polarized continuum model (PCM) [67,68] calculations at the M06-2X/6-311++G(2df,p) level of theory were performed. To this end, we calculated the free energy of solvation for all stationary points on the PES for a series of model isotropic solvents with varied polarities, including cyclopentane (ε = 1.96), 2-propanol (ε = 19.26), and dimethyl sulfoxide (ε = 46.83).

To calculate the standard state enthalpies of formation in the gas phase, Δ_f_*H*_m_^0^(g) (at *p*^0^ = 1 bar and T = 298.15 K), the explicitly correlated W1-F12 multi-level procedure [69] along with the atomization energy approach [69,70] was employed. Note that the conventional CCSD(T) calculations typically exhibit slow basis set convergence. This can be remarkably accelerated with the aid of the explicitly correlated F12 modifications [71]. Note that the W1-F12 approach used in this study has been slightly modified from the original version. More specifically, the B3LYP-D3BJ/def2-TZVPP optimized geometries were used (with the ZPE correction factor of 0.99) [72,73], and the diagonal Born-Oppenheimer corrections were omitted. The contributions from post-CCSD(T) excitations to the valence term of the atomization energies were controlled using the %TAE[(T)] diagnostics (viz., the percentage of the perturbative triples (T) in the CCSD(T) atomization energy) [74]. In the present study, these values did not exceed 5%, which supports the reliability of the CCSD(T)-F12 atomization energies calculated using the W1-F12 approach. The latter calculations were performed by the Molpro 2010 software [55]. The heats of formation at 0 K for the elements in the gas phase Δ_f_*H*_m_^0K^(g) [H] = 216.04 kJ mol^−1^, Δ_f_*H*_m_^0K^(g) [C] = 711.19 kJ mol^−1^, Δ_f_*H*_m_^0K^(g) [N] = 470.82 kJ mol^−1^, and Δ_f_*H*_m_^0K^(g) [O] = 246.79 kJ mol^−1^ were taken from the NIST-JANAF tables [75].

## 4. Conclusions

In the present work, we studied thermochemistry, tautomerism, and thermal stability of 5,7-dinitrobenzotriazole (**DBT**) and 4-amino-5,7-dinitrobenzotriazole (**ADBT**) using several complementary thermal analysis techniques and high-level quantum chemical calculations. Thermoanalytical experiments under elevated pressure allowed tracking of the thermolysis of **DBT** in the melt free of the concomitant vaporization. Model-fitting thermokinetic analysis shows that the kinetic scheme with two global consecutive steps provides the best fit for **DBT** decomposition. The first global stage (I) is a strong autocatalytic process that includes the noncatalytic (*E*_a1_^I^ = 173.9 ± 0.9 kJ mol^−1^, *log*(*A*_1_^I^/s^−1^) = 12.82 ± 0.09) and catalytic reaction of the second order with *E*_a2_^I^ = 136.5 ± 0.8 kJ mol^−1^, *log*(*A*_2_^I^/s^−1^) = 11.04 ± 0.07. The second global stage (II) is of non-integer order and corresponds to the secondary processes with the effective kinetic parameters *E*_a_^II^ = 134.5 ± 1.8 kJ mol^−1^, log(*A*^II^/s^−1^) = 8.7 ± 0.2.

Mechanistic insights into the decomposition process were obtained using quantum chemical calculations. We show that the *1H-5,7-tautomer* is a preferable form for both benzotriazoles in the gas phase and solutions. Theoretical calculations reveal a similar primary decomposition mechanism for **DBT** and **ADBT**. Namely, the energetically preferable channel is the nitro-nitrite rearrangement, with barriers of 267 and 276 kJ mol^−1^ for **DBT** and **ADBT**, respectively. However, due to significantly higher activation entropies (or, equivalently, preexponential factors), the C–NO_2_ cleavage (with the reaction enthalpies of 298 and 320 kJ mol^−1^, correspondingly) becomes the dominant channel at the experimental temperatures.

## Figures and Tables

**Figure 1 ijms-24-05330-f001:**
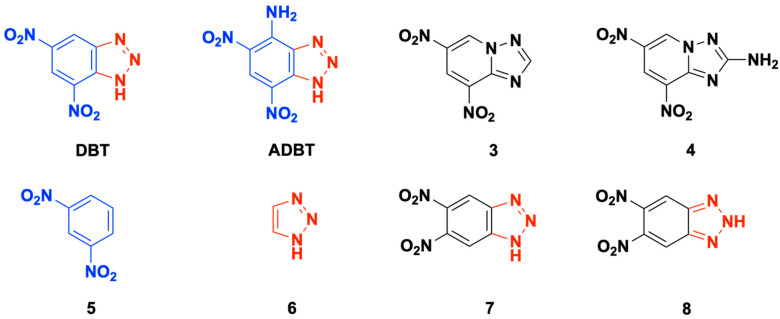
Molecular structure of 5,7-dinitrobezotriazole (**DBT**), 4-amino-5,7-dinitrobezotriazole (**ADBT**), and its structural isomers 6,8-dinitrotriazolopyridine (**3**) and 2-amino-6,8-dinitrotriazolopyridine (**4**), their simpler congeners 1,3-dinitrobenzene (**5**) and 1,2,3-triazole (**6**), and the tautomeric form of 5,6-dinitrobenzotriazole *1H*- (**7**) and *2H-form* (**8**). The dinitrobenzene part of the **DBT** and **ADBT** is colored blue, and the triazole one is red.

**Figure 2 ijms-24-05330-f002:**
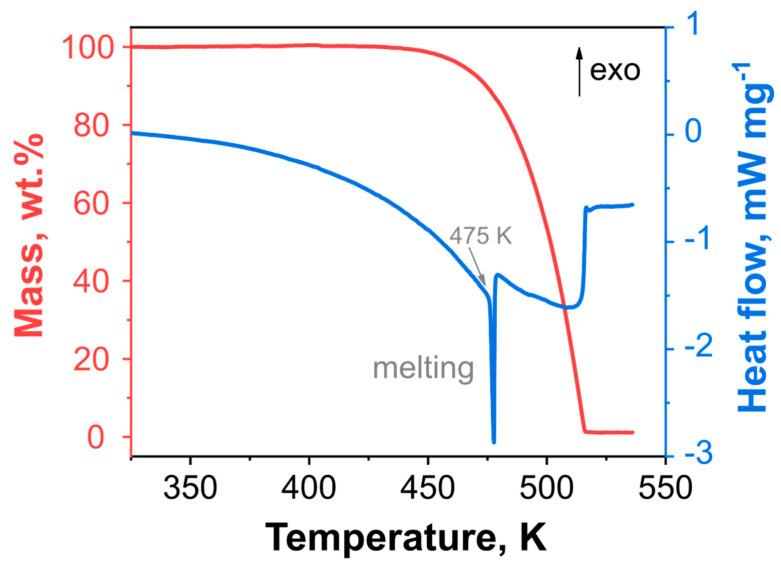
Thermal behavior of DBT under atmospheric pressure at the heating rate of 1 K min^−1^: the red trace is a thermogravimetry (TGA) curve, and the blue is the atmospheric pressure differential scanning calorimetry (DSC) curve, correspondingly.

**Figure 3 ijms-24-05330-f003:**
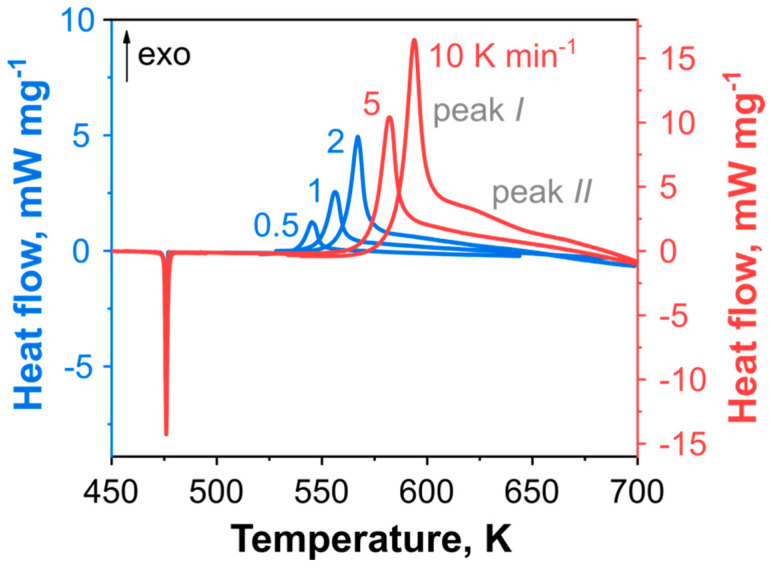
The DSC traces for linearly heated DBT at an elevated pressure of 2.0 MPa. Note the different Y-scale for low- and high-heating rate data.

**Figure 4 ijms-24-05330-f004:**
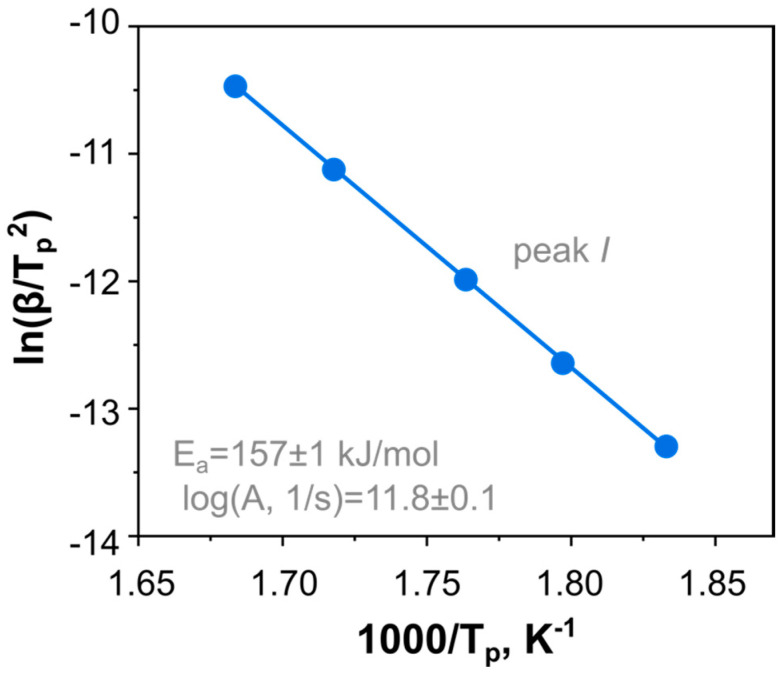
The Kissinger plot for the main peak (*I*) of DBT thermolysis under an elevated pressure of 2.0 MPa.

**Figure 5 ijms-24-05330-f005:**
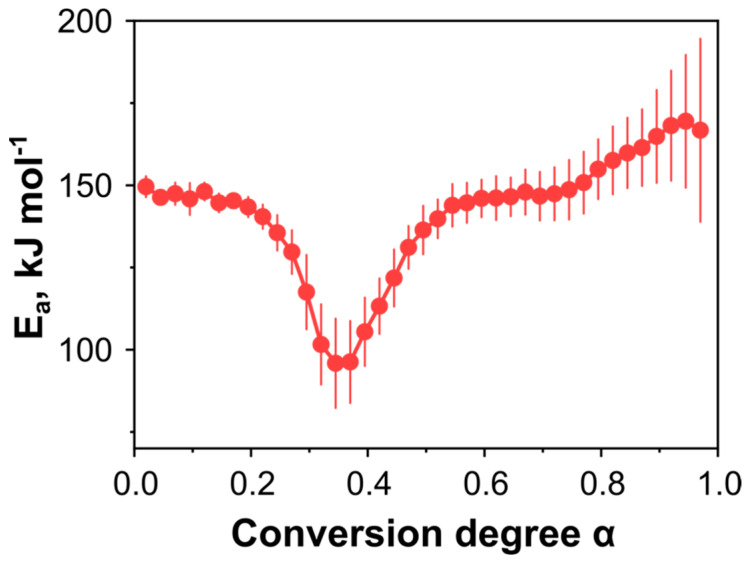
The apparent activation energy of the thermolysis of DBT was obtained by the isoconversional Friedman analysis from the DSC data under an elevated pressure of 2.0 MPa.

**Figure 6 ijms-24-05330-f006:**
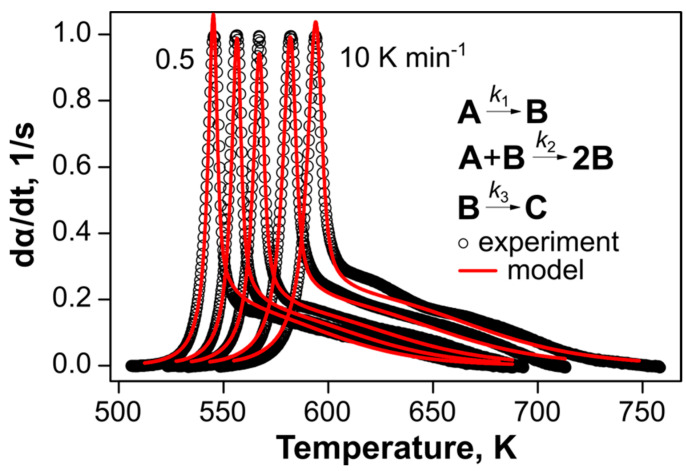
Normalized DSC data for **DBT** (the empty circles are rarefied experimental points) is fitted by the kinetic model (red curves). From left to right, the datapoints correspond to heating rates of 0.5, 1, 2, 5, and 10 K min^−1^.

**Figure 7 ijms-24-05330-f007:**
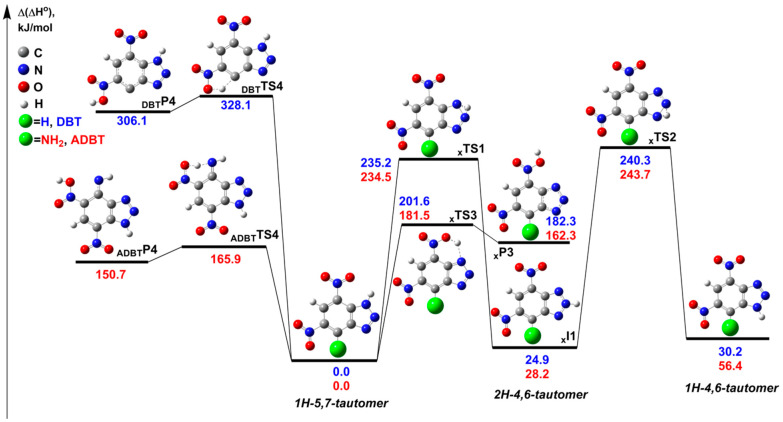
The relevant stationary points on the PES corresponding to the unimolecular tautomeric interconversions of **DBT** and **ADBT**. The tautomers that correspond to the crystalline structures of **DBT** and **ADBT** are chosen as the reference compounds for the calculation of the relative enthalpies at 298 K (Δ(Δ*H*°)). The geometry optimization and calculation of the thermal corrections were performed at the M06-2X/6-311++G(2df,p) level of theory, and the single point energies were computed at the DLPNO-CCSD(T)/jun-VQZ level. All energy values are given in kJ mol^−1^.

**Figure 8 ijms-24-05330-f008:**
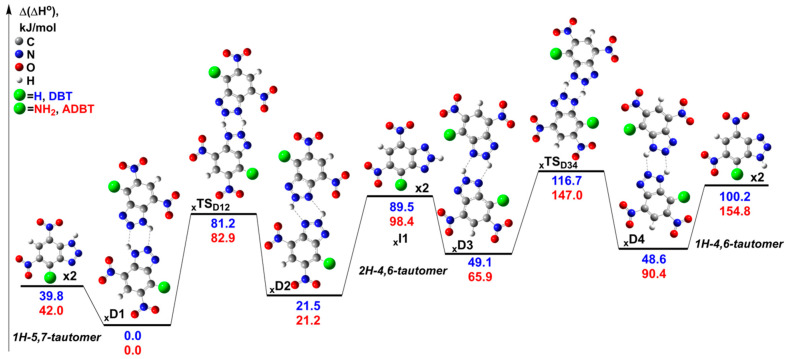
The relevant stationary points on the PES corresponding to the tautomeric interconversions of **DBT** and **ADBT** in dimers. The 1H-5,7-tautomers of **DBT** and **ADBT** are chosen as the reference compounds for the calculation of the relative enthalpies at 298 K (Δ(Δ*H*°)). The geometry optimization and calculation of the thermal corrections were performed at the M06-2X/6-311++G(2df,p) level of theory, and the single point energies were computed at the DLPNO-CCSD(T)/jun-VQZ level. All energy values are given in kJ mol^−1^.

**Figure 9 ijms-24-05330-f009:**
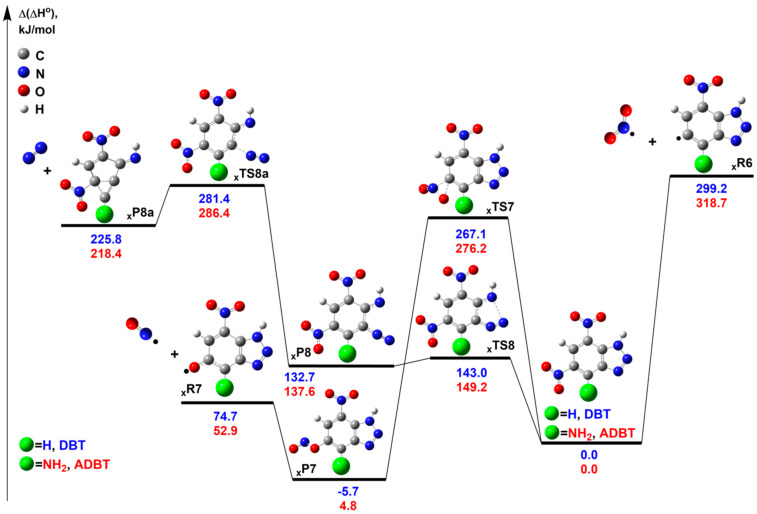
The relevant stationary points on the PES corresponding to the suggested decomposition pathways of **DBT** and **ADBT**. **DBT** and **ADBT** are used as the reference compounds for the calculation of the relative enthalpies at 298 K (Δ(Δ*H*°)). The most favorable decomposition channels are shown for the reactions **_x_R6** and **_x_TS7**. The geometry optimization and calculation of the thermal corrections were performed at the M06-2X/6-311++G(2df,p) level of theory, and the single point energies were computed at the DLPNO-CCSD(T)/jun-VQZ level. All energy values are given in kJ mol^−1^.

**Figure 10 ijms-24-05330-f010:**
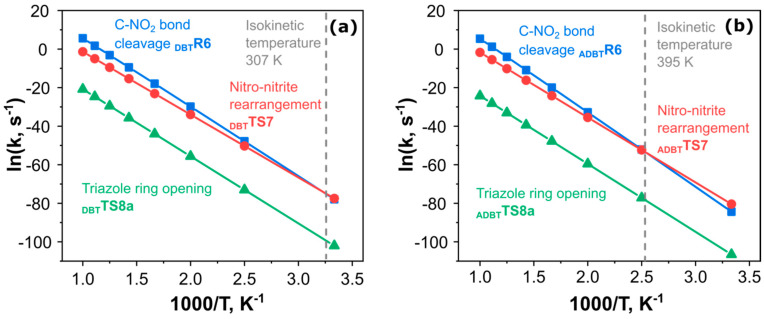
The Arrhenius plots for competitive reactions in the initial decomposition mechanisms for **DBT** (**a**) and **ADBT** (**b**): **xR6**–the C-NO_2_ radical cleavage bond (blue lines), **_x_TS7**–the nitro-nitrite isomerization (red lines), and **xTS8**–the triazole ring opening (green lines).

**Table 1 ijms-24-05330-t001:** The kinetic parameters calculated using the different approaches for the non-isothermal PDSC data of **DBT** decomposition.

Model Name	A	B	C	D	E
Analysis method	Kissinger	model-fitting	model-fitting	model-fitting	model-fitting
Kinetic model	–	ePT + ePT ^a^	ePT → ePT	AC ^b^ → ePT	AC → ePT
*log (A*_1_/s^−1^)	11.8 ± 0.1	12.61 ± 0.02	12.56 ± 0.02	18.0 ± 0.4/ 11.15 ± 0.07	12.82 ± 0.09/ 11.04 ± 0.07
*E*_a1_, kJ mol^−1^	157 ± 1	155.8 ± 0.2	153.7 ± 0.3	241.9 ± 4.8/ 141.2 ± 0.8	173.9 ± 0.9/ 136.5 ± 0.8
*n* _1_	–	1.26 ± 0.03	0.99 ± 0.03	1/1	1/1
*m* _1_	–	1.046 ± 0.006	1.95 ± 0.02	0/1	0/2
*q* _1_	–	0.999 ^c^	0.9	–	–
*log (A*_2_/s^−1^)	–	8.6 ± 0.1	5.6 ± 0.2	9.7 ± 0.2	8.7 ± 0.2
*E*_a2_, kJ mol^−1^	–	126.7 ± 1.6	90.9 ± 2.0	144.4 ± 2.3	134.5 ± 1.8
*n* _2_	–	3.00 ± 0.07	1.28 ± 0.05	2.50 ± 0.07	1.98 ± 0.05
*m* _2_	–	0.35 ± 0.02	−0.31 ± 0.01	−0.25 ± 0.04	−0.78 ± 0.06
*w*_1_ ^d^	–	0.300 ± 0.003	0.284 ± 0.004	0.347 ± 0.005	0.274 ± 0.007
*q* _2_	–	0.999 ^c^	0.9	0.999	0.999
*BIC* ^e^	–	−40,489.3	−43,590.5	−41,524.4	−45,643.4

^a^ The kinetic model denoted “ePT” corresponds to the extended Prout-Tompkins Equation (6) with the different connections between reactions, viz., “+”—independent parallel reactions and “→”—the consecutive reactions. ^b^ The kinetic model denoted “AC” corresponds to the conventional autocatalytic model with an *m*-th order by autocatalyst according to Equation (1). ^c^ These values were fixed during kinetic modeling. ^d^ The contribution of the first reaction to the total heat released is denoted as *w*. ^e^ The Bayesian informational criteria (BIC) values.

**Table 2 ijms-24-05330-t002:** The calculated kinetic parameters of the primary gas-phase decomposition reactions for **DBT** and **ADBT**. The rate-limiting transition state is shown in parentheses.

Reaction	Δ*H*°, kJ mol^−1^	*E*_a_, kJ mol^−1^	log(*A*/s^−1^)
**DBT** **→•_DBT_R6+•NO_2_**	299.2 ^a^	297.6	18.10
**DBT** **→ _DBT_P7 (_DBT_TS7)**	267.1	271.2	13.61
**DBT** **→ _DBT_P8 (_DBT_TS8a)**	281.4	289.4	21.10
**ADBT** **→•_ADBT_R6+•NO_2_**	318.7 ^a^	319.8	19.11
**ADBT** **→ _ADBT_P7 (_ADBT_TS7)**	276.2	280.3	13.88
**ADBT** **→ _ADBT_P8 (_ADBT_TS8a)**	286.4	293.3	20.14

^a^ In the case of barrierless reactions, the reaction enthalpies are shown.

**Table 3 ijms-24-05330-t003:** Thermal stability of the dinitrobenzotriazoles and dinitrotriazolopyridines: the effective rate constants of C–NO_2_ bond scission.

Species	DBT	ADBT	3	4
*k_eff_*(650 K) ^a^	1	0.1	18.3	5.5

^a^ The relative rate constants with respect to that of **DBT** are shown.

**Table 4 ijms-24-05330-t004:** The enthalpies of formation in the gas phase ($\Delta_{f}H_{m}^{{}^{\circ}}(g)$) and solid state ($\Delta_{f}H_{m}^{{}^{\circ}}(s)$) along with the sublimation enthalpies ($\Delta_{s}^{g}H_{m}^{{}^{\circ}}$) for dinitrobenzotriazoles (**DBT** and **ADBT**) and relative dinitrotriazolopyridines (**3** and **4**).

Compound	$\Delta_{f}H_{m}^{{}^{\circ}}(g)$, kJ/mol	$\Delta_{s}^{g}H_{m}^{{}^{\circ}}$, kJ/mol	$\Delta_{f}H_{m}^{{}^{\circ}}(s)$, kJ/mol
**DBT**	308.1 ^a^	121.8 ± 3.2 120	187.3 179 ^c^
**ADBT**	271.2 ^a^	150	129.5 128 ^c^
**3**	308.6 ^a^ 262.3 ^b^	124	184.4 191.6 ^d^
**4**	279.4 ^a^	150	129.6 138.1 ^d^

^a^ Calculated using the W1-F12 multi-level procedure and atomization energy approach. ^b^ The gas phase enthalpy of formation is calculated using the G3B3 multi-level procedure and atomization energy approach [43]. ^c^ Calculated as a combination of the gas-phase formation enthalpy from the CBS-4M multi-level procedure and the sublimation enthalpy estimated by the Trouton rule [16]. ^d^ Calculated as a combination of the gas-phase formation enthalpy from the G3B3 multi-level procedure and the sublimation enthalpy estimated by the Trouton rule [44].

## Data Availability

Data are contained within the article or Appendix A.

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
