# Peer review of "Thermochemistry, Tautomerism, and Thermal Stability of 5,7-Dinitrobenzotriazoles"

_ijms, 2023, doi:10.3390/ijms24065330_

Round 1
Reviewer 1 Report
Triazoles MDPI 22=23
Two minor comments -- only
Table 2 – further clarify ( more clear definitions) the specific property of the three columns of data
Ln 404 reactions between the dimers ?-- clarify
Author Response
Reviewer 1
We thank the referee for positive comments and helpful suggestions.
Comment: Table 2 – further clarify (more clear definitions) the specific property of the three columns of data
Response: We have clarified the header and column titles, they are highlighted in the text.
Comment: Ln 404 reactions between the dimers ?-- clarify
Response: We have considered the dimers, not tetramers, because the most important effect is already present in the former species – that is, a significant reduction of the activation barriers of H-transfer reactions in the different tautomers. Once these reactions are faster than the monomolecular decomposition channels, the tautomer equilibrium is maintained in the course of thermolysis.
Reviewer 2 Report
Review is attached as a PDF file.

Author Response
Reviewer 2
We are grateful to Reviewer 2 for their thorough reading of our manuscript and valid and helpful comments. His\her suggestions helped us to look at our paper critically. We have considered all of the concerns while revising our manuscript.
Comment: If one decides to tune the properties of organics by substitution why not to use the palette of substituents and study all features systematically (Hammett)?
Response: We thank the referee for this point. This would be indeed an interesting separate study. Here we concentrated on the properties of the available energetic materials. As we emphasized in the Introduction:
“DBT melts at 466 K (193°Ð¡) and then decomposes with the extrapolated onset of 564 K (291°C), while ADBT melts with a subsequent decomposition at 610 K (337°C) [16]. These values are the only available information on the thermal behavior of DBT and ADBT.”
Thus, we do believe the present manuscript to contribute sufficiently to the understanding of the thermal stability of the titled species.
Comment: In the abstract is should be understanding their thermal stability.
Response: We have clarified this in the abstract:
“Theory suggests similar decomposition mechanisms for DBT and ADBT with the most favorable channels being nitro-nitrite isomerization and C–NO2 bond cleavage. … At the same time, due to the higher preexponential factor, the radical bond cleavage with the reaction enthalpies 298 and 320 kJ mol-1, correspondingly, dominates in the experimental temperature range. In line with the theoretical predictions of C–NO2 bond energies, DBT is more thermally stable than ADBT.”
Comment: I do not see the reason to compare compounds 4 and ADBT in the same data set. Those are only similar by composition but in ADBT the intramolecular hydrogen bond stabilizes its structure.
Response: We thank the referee for a good point for discussion. Yes, the intramolecular H-bonds are present in the crystal structure in ADBT. However, we aimed at considering the “intrinsic” decomposition properties of ABDT and 4 and this was one of the reasons for incorporating the computational data to the manuscript. As seen from Table 3 and a discussion therein, the kinetic stability of an isolated ABDT species is higher as well.
Comment: Fig. 7 shows the tautomeric states of DBT and ADBT and transition states related to proton transfer reaction. The structure labelled as xP3 is an aci-nitro form with the relatively high energy. That fact is easy to understood as a) the lone electron pairs of nitrogen and oxygen atoms exhibit repulsion and b) there is a lack of hydrogen bonding that, according to Etter’s rules, is formed. Thus, in my opinion, the quality of described PES is doubtful.
Response: The aci nitro isomers are generally higher in energy than their nitro counterparts, both for aliphatic and aromatic compounds. This fact is well documented in our previous works, e.g., 10.1016/j.tca.2020.178697, 10.1039/d1cp04663b, 10.1021/jp906853e. This fact confirms the reliability of the described PES. Apart from this, the latter is calculated using the high-level post-HF methods (viz., DLPNO-CCSD(T)), whose predictive ability is far beyond the conventional DFT. Finally, whatever the case, the aci-forms do not play an important role in the decomposition mechanism. This fact agrees well with all above mentioned refs, too.
Comment: Fig. 8 shows the intermolecular interactions in considered tautomers. Unfortunately, the mixed-model was not taken into account, which means, for example, the 1H-5,7-tautomer (Fig. 7) to be considered as doubly hydrogen bonded complex with 2H-4,6-tautomer an remaining one. When the sample (experiment) is heated not only symmetric hydrogen bonding takes place. While the aim of the research is to study the decomposition and its mechanism those additional structures should be seriously considered.
Response: The referee is generally correct. However, the most important effect is already present in the symmetric dimers – that is, a significant reduction of the activation barriers of H-transfer reactions in the different tautomers. Once these reactions are faster than the monomolecular decomposition channels, the tautomer equilibrium is maintained in the course of thermolysis. Any possible further reduction of the barriers does not add up anything significant kinetically. Moreover, the reactions of all other tautomers turned out to be unimportant as well.
Comment: The decomposition was considered as an autocatalytic process. How was it reflected DFT calculations?
Response: This is also a good point. We have considered a variety of possible secondary autocatalytic reactions of similar to some extent energetic materials in our recent work (ref 72 in a manuscript). There we found a rather unexpected result – namely, a heterocyclic primary radical without NO2 is a key catalytic species for secondary reactions, not NO2. Here even a kinetic kinetic equation (4) exhibits a second order with respect to a product. Thus, we decided to suspend a computational discussion at a particular point.
Comment: The proposed mechanism shown in Fig. 9 is, in part, convincing. The ring opening is below the TS related to proton transfer shown in Fig. 7. On the other hand, the barrier ca. 140 kJ/mol is higher than double proton transfer in symmetric dimer xTSD12 shown in Fig. 8.
Response: Yes, indeed, this is correct. As we discussed above, this only means if there were any lower effective channels, we’d be important. Finally, we found that only the lowest tautomeric form is important.
Comment: I doubt the protocol used for reactivity in the melt is correct. Why the three solvents were used? What is the add-on from that approach? Those solvents were chosen as a) non-polar, b) carrying hydrogen bond donor and c) carrying hydrogen bond acceptor and those differ very much by polarity. There is no clear message from this approach and the Fig. S8 does not tell much except for sophisticated decomposition path in solution based on theoretical considerations and calculations.
Response: We used the three solvents with polarity varied in a wide range to estimate the possible changes of the decomposition mechanism of DBT and ADBT in the melt, since their polarities are generally unknown. PCM calculations are a simple and efficiency way of estimating how the gas-phase reactions are changed upon the transition to a solvent. We found that the decomposition mechanism of DBT and ADBT does not has a significant sensitivity to polarity of solvents. Specific properties like H-bonding were partly considered separately in explicit dimer calculations. Generally, this procedure yields a good approximation to what happens in a melt. Any very tiny details cannot be caught, this is right. However, at a given level of accuracy this is indeed reasonable, see 10.1016/j.tca.2020.178697, 10.1039/d1cp04663b
Comment: At the end let me express my feeling about the submitted manuscript. It looks like being two parts joined together but not answering fundamental questions. I think that division of this material for a) purely experimental one and b) purely theoretical inclusing solvent effects, would be much better for understanding decomposition of the subjected molecules in solid AND in solution .
Reply: We do not agree that the “two parts joined together but not answering fundamental questions”. On the contrary, we clearly found a molecular-level confirmation for a particular channel among many possible elemetary reactions generally believed to occur in the literature. We indeed agree that we did not build a comprehensive detailed kinetic model with all hundreds of elementary reactions occurring in the melt and the gas phase leading from the pristine samples to simple products. However, we believe the reported bunch of results to add significantly to this field. Note that even in the case of extremely well studied RDX and HMX these mechanisms have not been built so far.
Reviewer 3 Report
The study carried out by the authors deals with the thermochemistry of nitro-substituted benzotriazoles. The behaviors of two compounds (DBT and ADBT) have been observed under heating. The authors have used various methods to explore that thermolysis: pressure DSC, advanced thermokinetic methods (both isoconversional and model-fitting approaches) and high-level quantum chemical calculations. Their work on DBT and ADBT was compared with that performed on other known molecules bearing similar fragments (dinitrotriazolopyridine and aminodinitrotriazolopyridine) and with simpler structures (1,3-dinitrobenzene and 1,2,3-triazole). Their thermal stability was studied regarding its kinetics and its decomposition mechanism. The tautomerism of the compounds was explored in terms of stability and interconversion mechanism.
This work is pure research, far from direct applications, and perhaps of niche interest.
The paper is well written (despite a few details needing minor corrections), the study thorough, methodically led and abundantly supported by external references, the results novel and sound.
l. 13: remove “the” before “understanding”
l. 19: add “a” before “kinetic scheme”
l. 40: change “With” for “While”
l. 44: remove “was” before “confirmed”
l. 65: remove “to” before “be”
It is worth publication as it is (after those cosmetic syntax corrections).
Author Response
Reviewer 3
We thank Reviewer 3 for his/her time and positive assessment of our work.
Comments:
- 13: remove “the” before “understanding”
- 19: add “a” before “kinetic scheme”
- 40: change “With” for “While”
- 44: remove “was” before “confirmed”
- 65: remove “to” before “be”
Response: We have corrected all noted issues and typos. We have also edited the whole manuscript once again.